# Key priority areas for patient safety improvement strategy in Libya: a protocol for a modified Delphi study

Mustafa Elmontsri, Ricky Banarsee, Azeem Majeed

## ABSTRACT

**Introduction** Patient safety is a global public health problem. Estimates and size of the problem of patient safety in low-income and developing countries are scarce. A systems approach is needed for ensuring that patients are protected from harm while receiving care. The primary objective of this study will be to use a consensus-based approach to identify the key priority areas for patient safety improvement in Libya as a developing country.

**Design** A modified Delphi study.

**Methods and analysis** A three-phase modified Delphi study will be conducted using an anonymous web-based questionnaires. 15 international experts in the field of patient safety will be recruited to prioritise areas of patient safety that are vital to developing countries such as Libya. The participants will be given the opportunity to rank a list of elements on five criteria. The participants will also be asked to list five barriers that they believe hinder the implementation of patient safety systems. Descriptive statistics will be used to evaluate consensus agreement, including percentage agreement and coefficient of variation. Kendall's coefficient of concordance will be used to evaluate consensus across all participants.

**Ethics and dissemination** Ethical approval has been granted from Imperial College Research Ethics Committee (ICREC: 16IC3598). The findings of the study will be published in a PhD thesis. A manuscript will also be prepared for publication in a high-impact peer-reviewed journal describing the Delphi process and the findings of the study.

## Strengths and limitations of this study

► This modified Delphi study is the first study that focuses on the issue of patient safety in Libya using this methodology.
► The Delphi will recruit international experts in the field of patient safety from regional and global organisations to reflect on their experiences in this field.
► The modified Delphi method will minimise the domination of individual participants through anonymity, which will help in eliminating group pressures for conformity.
► Due to lack of patient safety strategy in Libya, this protocol builds on existing research and data from other high-income countries.
► The modified Delphi study aims to include 15 international experts, which might limit the generalisability and validity of the findings.

Department of Primary Care and Public Health, School of Public Health, Imperial College London, London, UK

**Correspondence to**
Mustafa Elmontsri; m.elmontsri10@imperial.ac.uk

## INTRODUCTION

In the last decade, considerable efforts have been made to improve the safety of healthcare in which it contributed to the widespread acceptance and awareness of the problem of medical harm.[1] In many countries, major progress has been made in assessing the scale and nature of harm.[2] Several studies have examined the extent to which patients are harmed while receiving medical care, for example, the nature and scale of adverse drug events, surgical adverse events, infection and medication prescriptions have been catalogued.[1] Reliable indicators of safety status have been developed over the past decade, but the measurement and monitoring of safety continuous to be a challenge.[3] A wide range of contributory factors have been revealed following the analysis of safety incidents. Patient safety incidents in high-income countries receive huge attention from the public and the government in which public enquires are ordered to identify the root causes. A recent inquiry in the UK revealed that the substandard performance of individuals due to incapacity or sickness are linked to patient safety problems in the workplace and led to the occurrence of harm to patients.[4] However, other causes include 'a culture focused on doing the system's business—not that of the patients; too great a degree of tolerance of poor standards and of risk to patients; a failure of communication between the many agencies to share their knowledge of concerns' were contributory factors to poor patient safety performance.[4] In many high-income countries, individuals and organisations are being increasingly regulated as well as encouraged to report consistent poor performance to help learn lessons for improvement. The Institute of Medicine report 'Crossing the quality chasm: a new health system for

the 21st century',[5] indicated that large gaps between the care patients should receive and what they receive exist and recommended the need to search for a new system design to improve performance. Over the past decade, frameworks and models that organise and stimulate the development of theory and practices within the field of patient safety have been proposed. One of the models that encourage a systems approach to patient safety is the System Engineering Initiative for Patient Safety (SEIPS) model.[6] The SEIPS model was constructed based on health systems in high-income countries and puts much emphasis on the role of technology as part of the work system components. The SEIPS model emphasised on the need to have an integrated approach to patient safety based on the most well-known model of healthcare quality of Donabedian known as Structure, Process and Outcomes model.[7] However, in order for patient safety to be improved, action should be taken at all levels: global, national and local.[8] The global level should focus on sharing knowledge, recommendations and standards across national and local organisations, whereas the national level should focus on the development of health system attributes and policies such as the introduction of regulations and coordination of health policies towards improving patient safety. Finally, the local level should execute the interventions proposed to ensure they are relevant and effective.

### Patient safety in low-income and developing countries

Countries at all levels of development are affected by patient safety as it is a global public health issue which is also a fundamental requirement in healthcare delivery.[9] Estimates of the size of the patient safety problem in high-income countries have been identified by several studies,[10 11] whereas in low-income and developing and transitional countries such estimates are scarce. Among the patient safety risks that are present in low-income and developing countries include hospital-acquired infections, surgical complications, delays in diagnosis and adverse drug reactions.[12] Similarly, Harrison et al[13] found that medication errors, patient infection and poor maternal and perinatal care are among the common patient safety risks in low-income and developing countries in Southeast Asia. WHO estimated that millions of patients suffer disabling injuries or death due to unsafe medical care every year.[14] There is very little known about the scale of unsafe care in low-income and developing countries when compared with high-income nations. A large-scale study was conducted by the WHO Patient Safety arm from 2006 to 2008 to examine the extent of harm in eight countries from the Eastern Mediterranean and Africa in which 26 hospitals took part in the study. A total of 18 000 patients' experiences were examined in these two regions and the study found that 8 in 100 of the patient studies were exposed to health care-related harmful incident.[12] The study also found that among the factors which contribute to adverse events in low-income and developing countries include: inadequate

training or supervision of clinical staff, no protocol, no policy or failure to implement, inadequate communication or report and inadequate staffing. There have been infrequent and limited scope of patient safety research in low-income and developing and emerging countries.[15] The WHO World Alliance for Patient Safety was created as a result of the alarming issue of patient safety, especially in low-income and developing countries. WHO has also requested all member states to 'pay the closest possible attention to the problem of patient safety'.[16] It is also recognised by WHO that patient safety improvement in low-income and developing countries require different strategies due the limited capacity and infrastructure of these countries. Such strategies would need to be developed based on the available resources and expertise while taking into account the lack of regulations and accountability within such health systems.

### Aims

The aim of this Delphi study is to identify key priority areas for patient safety improvement in low-income and developing countries and countries in transition. The study aims to achieve consensus on the most important areas to improve patient safety in low-income and developing countries while using Libya as a case study. The study objectives include the following:

► Identify the most important areas in regard to improving patient safety in low-income and developing countries.
► To seek expert advice to identify the key factors that are the most important in the improvement of patient safety.
► To identify the barriers that hinder the implementation of patient safety systems.
► To develop a patient safety improvement model that is oriented to the Libyan context and capable of improving patient safety in the Libyan healthcare system.

Libya is chosen as a case study as the country is undergoing a transitional stage. Libya provides universal health coverage free of charge. The health system in Libya is mainly funded by the state. Health services are delivered through primary healthcare centres, polyclinics, rehabilitation centres and general hospitals in urban and rural areas.

### METHODS AND ANALYSIS

A modified Delphi approach will be adopted for this study to identify the key priority areas for patient safety improvement in Libya. This modified Delphi adopts a structured questionnaire for round 1 and a small sample size.[17] The Delphi technique has been widely used in health services and policy research as it supports decision-making and gives a way of structuring a large mass of information and expertise so that informed judgement, forecasting and decision-making can be achieved.[18] It can be used to discuss both numerical and non-quantifiable nature of

problems. It is believed that the Delphi technique has the capacity to deal with ambiguity and multidimensionality based on its ability to draw on the informed judgement of a group of experts systematically, which made it one of the highly reliable tools to support decision-making in the fields of medicine, sociology and policy making.[19–21] A key strength of the Delphi technique is the fact that it does not force consensus, but helps in identifying where agreement exists among international experts.[22–24] Some of the many advantages of the Delphi technique are: ability to conduct a study in geographically dispersed locations as it is the case of this protocol, time and cost-effectiveness as well as allows the discussion of complex and broad problems.

### Use of Delphi method in patient safety research

The Delphi method was used to identify the barriers to implementation of patient safety systems in healthcare institutions with 23 experts in healthcare quality and healthcare systems approach in which 18 US states were represented.[25] The Delphi method was also used to determine developmental progression of quality and safety competencies in nursing education with 18 subject matter experts.[26] It was also used by the WHO World Alliance of Patient Safety to develop an International Classification for Patient Safety in which 253 experts responded to the first round of the Delphi method.[22] The Delphi method was also used to identify and assess for face and content validity of a group of safety indicators with a panel of 30 experts representing all specialities working in labour and delivery units.[27] It was also used by researchers in the USA to identify priority patient safety outcome measures for use in monitoring and evaluating progress in improving patient safety with 47 national clinical and research experts.[28] The Delphi method was used to identify the causes of patient safety incidents and devise solutions for patient safety in primary care in which 20 physicians and patient safety experts were recruited.[23] A three-round Delphi study was also carried out with a panel of 20 national experts in Canada to develop a set of Canadian consensus-based indicators for the safe use of medication for both inpatient and outpatient settings.[29]

Three rounds of the Delphi method will take place as shown in figure 1; each round will be distributed by email.

### Delphi method rounds

► Pre-Delphi process: an email invitation will be sent to the panel of international experts requesting their voluntary participation in the Delphi study. The experts come from different geographical areas

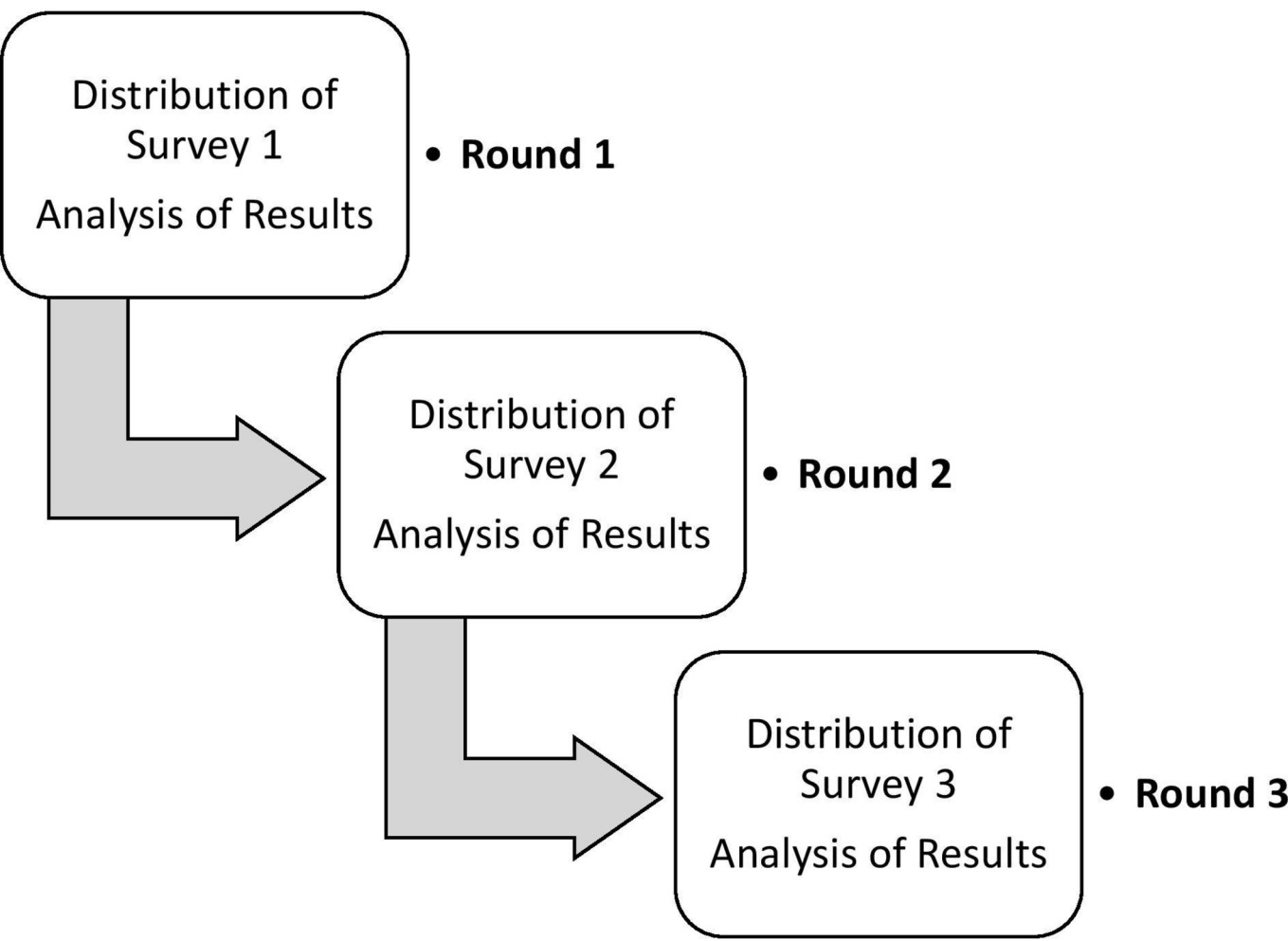

**Figure 1** Delphi method process.

including those from North Africa and the Middle East, WHO and from the UK and the USA. The experts are selected based on their familiarity with health systems in low-income and developing countries as well as their wealth of publications in the field of patient safety. Those who agree to participate will be sent a web-based questionnaire along with a consent form and participant information sheet providing more details about the purpose of the Delphi study.

► Round 1: all panel members will be asked to rank a number of elements based on the preprepared identified domains and topics using an online questionnaire. As part of round 1, the panel members will be asked to identify five barriers to implementation of patient safety systems in healthcare institutions.

Communication with participants via email will be according to the following schedule:
► Distribution of participant information sheet, consent form and survey link
► Reminder email 1 week later
► Reminder email 2 weeks later
► Round 2: each panel member will receive feedback of a list of themed topics with supporting statements. Panel members will be asked to rate the importance of each topic using a 5-point Likert scale. They will also be asked to rank the barriers based on the reported common barriers in round 1.

Communication with participants via email will be according to the following schedule:
► Distribution of the new survey link
► Reminder email 1 week later
► Reminder email 2 weeks later
► Round 3: each panel member will receive feedback in the form of a list of themed areas/topics/elements reaching consensus with their supporting statements. Each panel member will be asked to rate each area/topic/element again. Each panel member will be invited to make comments about the prioritisation process.

Communication with participants via email will be according to the following schedule:
► Distribution of the final survey link
► Reminder email 1 week later
► Reminder email 2 weeks later.

## Sample and location
Expert panel members will be included from different regions in the world including experts from the UK, the USA, Arab Countries, Africa and WHO. The panel members are chosen because of their extensive and invaluable input into the field of patient safety evidenced by their publications in the field. An email will be circulated to each member of the expert panel by the researcher inviting them to participate and respond to the researcher via email to confirm their willingness to participate in the study. Those who wish to participate will

be contacted by the researcher with the survey link and a copy of the participant information sheet and consent form. Each questionnaire round will be circulated via email to be completed in the privacy of the individual's home to avoid bias or coercion. All responses within the expert panel will be anonymised prior to collation and circulation.

## Analysis
Descriptive statistics will be used to evaluate consensus agreement, including percentage agreement and coefficient of variation. The latter is a normalised measure of dispersion of a probability distribution. The coefficient of variation (CV) is defined as the ratio of the SD to the mean. A level of consensus will also be established. Kendall's coefficient of concordance will be used to evaluate consensus across all participants. This is a non-parametric statistic. It can be used for assessing agreement among the participants. Kendall's $W$ ranges from 0 (no agreement) to 1 (complete agreement).

The analysis will include:
► Mean rating
► Percentage agreement
► Coefficient of variation
► Kendall's coefficient of concordance

Consensus will be established if:
► Mean rating of 3.5
► Percentage agreement of ≥75%
► Coefficient of variation of ≤20%
► A median ranking of 4 or more

## DISCUSSION
For patient safety to be truly global, and as highlighted by the Institute of Medicine, low-income and middle-income countries need to be included in the patient safety community.[30] Thus, it could be argued that over the coming 15 years, harm reductions are likely to come from these countries. More importantly, there is a need to ensure that those tools which have been developed in high-income countries can also be developed in low-income countries. Countries in transition such as Libya can also be fertile for innovation in the healthcare sector. There is also a need for the healthcare system in low-income and developing countries to have the right capabilities to produce changes in the delivery and design of services so that the quality and value of healthcare in the 21st century can be improved. Satisfaction, biological and functional parameters should be used to measure the outcomes as part of the redesign efforts of healthcare systems.[31] The capacity of the healthcare system should be built for improvement by successful healthcare leaders of the 21st century in low-income and developing countries. Policy makers and all stakeholders in healthcare organisations in low-income and developing countries should focus on the need of aligning all organisational strategies for improvement (professional development, financing, operations and

education) and should perform patient assessment, data gathering, process assessment and outcome measurement to achieve organisational improvement.[32] In other words, the professional environment, staff, patient and family as well as the leadership and management should all be engaged in all phases of the improvement work. For example, the management and leadership of healthcare organisations can play a significant role in improving patient safety through the establishment of caring relationship with clinical staff, building patient safety infrastructure and promotion of organisational culture change in supporting the quality and patient safety improvement efforts.[33 34] The absence of legal frameworks regarding patient safety improvement along with limited resources and expertise in the field could also influence on the capabilities of health systems in low-income and developing countries. Due to the limited resources, investment in information technology systems in low-income and developing countries may not be possible and that will impact on the overall improvement of care delivered to patients.

An agreed set of indicators that are suitable for international comparison which can also help in forming the basis of local measurement systems is highly needed as part of the global movement for patient safety. Global research on patient safety should shift the focus on investigating the issues that low-income countries and countries in transition face in improving their patient safety performance.[35] Research collaboration across countries and international organisations would be very beneficial for countries that have poor infrastructure for conducting empirical research studies. The findings of research could also be disseminated on a global and regional scale to help low-income countries translate these findings into policy recommendations. Patient safety requires a systems approach as it is highly dependent on systemic factors. Redesigning the healthcare system for safer care delivery requires a systems approach in which individual and organisational factors are taken into account.[36] Healthcare institutions would be vulnerable to mistakes in patient safety if a narrow focus on one isolated process without taking into consideration the systematic nature of medical errors is adopted.[37] Hence, this modified Delphi study will help in establishing a point of reference for policy makers in low-income and developing countries on the vital elements and components that are required when developing a national strategy for patient safety improvement. This research would also help in bridging the gaps between developed and developing nations and to encourage collaborative research projects between institutions and researchers so that sustainable solutions and improvements can be made.

## Ethics and dissemination

Ethical approval was granted by Imperial College Research Ethics Committee (ICREC: 16IC3598). Confidentiality and anonymity will be kept throughout the study phases and all individual responses will be anonymised prior to collation by the researchers involved in the analysis of this study. Only members of the research team will have access to the codes that will be used for coding the Delphi questionnaires. Data which will be published will not identify the identities or organisations of the individuals participated in the study. The findings of the study will be published in a PhD thesis. A manuscript will also be prepared for publication in a high-impact peer-reviewed journal describing the modified Delphi process and the findings of the study. We also expect that the findings of this study will be disseminated to national, international and regional audiences via conference presentations, posters and seminars. Participants in the Delphi study will also be forwarded a copy of the final report. Finally, it is expected that the findings of this study will initiate collaborative research between developing and developed nations about improving patient safety.

## Conclusion

All healthcare professionals and stakeholders in the process of healthcare delivery should have personal responsibility to patient safety. A positive culture of patient safety is highly needed in low-income and developing countries to ensure that discovering and reporting errors is being rewarded not punished. This modified Delphi study will help in identifying the key priorities for the improvement of patient safety in low-income and developing countries using Libya as a case study and will also help shifting the focus of policy makers in these countries on the key important elements that are needed when developing a national strategy for patient safety improvement. The study will identify the barriers that hinder the implementation of patient safety systems so that comprehensive and efficient national patient safety policies and strategies are developed.

**Contributors** ME and RB prepared the study design and the content of the study protocol. MA further improved the quality of the protocol with important intellectual revisions and assisted in the development of the survey questionnaire for round one. ME is responsible for drafting the protocol manuscript, which was then revised and approval by RB and MA.

**Funding** This study is part of ME's PhD research, which was funded by the Government of Libya. The Department of Primary Care and Public Health at Imperial College London is grateful for support from the NW London NIHR Collaboration for Leadership in Applied Health Research & Care (CLAHRC), the Imperial NIHR Biomedical Research Centre and the Imperial Centre for Patient Safety and Service Quality (CPSSQ).

**Competing interests** None declared.

**Ethics approval** Imperial College Research Ethics Committee ICREC: 16IC3598.

**Provenance and peer review** Not commissioned; externally peer reviewed.

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
