## [Reviewer comments · BMJ Open]

ARTICLE DETAILS

TITLE (PROVISIONAL)	Key Priority Areas for Patient Safety Improvement Strategy in Libya: A Protocol for a modified Delphi Study
AUTHORS	Elmontsri, Mustafa; Banarsee, Ricky; Majeed, Azeem

VERSION 1 - REVIEW

REVIEWER	Marise Reis de Freitas Universidade Federal do Rio Grande do Norte Brazil
REVIEW RETURNED	31-Oct-2016

GENERAL COMMENTS	This paper will contribute to research in developing countries. Please in line 33 at abstract, you have to substitute the word hard for harm.
---

REVIEWER	Helen Niemeyer (for Christine Knaevelsrud) Freie Universität Berlin, Germany
REVIEW RETURNED	30-Jan-2017

GENERAL COMMENTS	The protocol under review deals with a very important topic and has several strengths. However, a number of aspects require a revision. 1. p.1, highlight 1 is unclear: Does "looks at research in Libya" mean that research conducted in Libya is analysed?2. The english language of the protocol needs to be revised, e.g. first sentence on p. 2.3. What is the meaning of the sentence "A recent inquiry in the United Kingdom ... workplace." on p. 2 in the context of patient safety?4. Can the authors provide examples for the most important patient safety risks in developed countries?5. Will the study authors serve as oversight committee in the study?6. Does any of the authors have expertise on the Delphi method, e.g. serve as methodologist in former Delphi studies?7. Prior to this project, did the authors conduct reviews of guidelines and the existing literature on patient safety (e.g. in developed countries)? The authors briefly refer to the SEIPS model. Please provide sufficient information about the model, whether it is related to developed or developing countries, and whether questions were derived from it for the Delphi process?8. Were questions for the Delphi process derived from the WHO papers that the authors refer to, which seem to be specifically about developing countries?9. Can the authors provide reasons why Libya was chosen?10. Can the authors provide basic information about the current state of the Libyan health care system?
---

	11. One aim of the study (p.4) is to develop a model that is tailored to the Libyan context. Is this step supposed to be conducted by the international experts on patient safety? In that case, how do the authors make sure that their knowledge about Libya is sufficient? 12. Which time frame do the authors base their model for the implementation of patient safety structures in the Libyan health care system on (e.g. the next 10 years), in order to make it practically applicable? 13. Can the authors provide detailed information on the web-based questionnaire they apply in the pre-Delphi process and on the pre-prepared identified domains and topics in the online questionnaire in round 1 (p. 5)? 14. Does the step that the panel members have to identify five barriers to implementation have anything to do with the domains in the online questionnaire? Do the members have to rank the five most important domains, or are domains and barriers two different topics? 15. How are the experts defined? Can the authors provide more information on their profession and scientific background (p.6)?
--	---

REVIEWER	Professor Salman Rawaf Imperial College London UK I know the three authors as they work with me in the same department. However, I am not involved in this research (PhD Research topic and not Departmental Research)
REVIEW RETURNED	08-Feb-2017

GENERAL COMMENTS	This is an important protocol on topic go great interest and implications in developing countries. While the flow of the paper is good I believe the linguistic issues have to be addressed before any consideration for publication (please see my comments highlighted on the attache manuscript). The reviewer also provided a marked copy with additional comments. Please contact the publisher for full details.
--

REVIEWER	A-Prof Guy Haller Geneva University Hospitals-Switzerland
REVIEW RETURNED	26-Feb-2017

GENERAL COMMENTS	This protocol describes an expert consensus gaining study based on the Delphi methodology. It aims at identifying key factors that are the most important in the improvement of patient safety and at highlighting barriers that may hinder the implementation of these factors in developing countries. Libya will be used as a case-study. The study protocol plans to recruit experts in Africa, the World Health Organization, the United States, the United Kingdom and Arab countries. Such studies are important to the overall advancement of the patient safety agenda across countries and more specifically, in developing countries. While the study is important and the protocol includes detailed information regarding study aim and gap in knowledge, the lack of a clear and comprehensive methodology section in the manuscript hinders the protocol to reach its goal.
--

	Additional details are needed at this stage. First, there appears to be some tension between the concept of an extensive Delphi process aimed at identifying needs in developing countries and the choice of Libya as a case-study. It is unclear how factors identified on Libya could be transposed to, for instance, Guatemala, Cambodia or the Philippines, all being developing countries. The generalizability issue is substantial in this study. Secondly, a Delphi study is not a qualitative study. As a result, detailed information and rating sheet that will be sent to expert should be provided. This sheet should include all factors and aspects on which expert opinion will be sought. Thirdly, it is unclear how a highly selected group of experts in Africa, the World Health Organization, the United States, the United Kingdom and Arab countries are representative of all developing countries. What about Asia, South America, the Pacific islands? Finally, the statistical method provided in the protocol does not relate to a Delphi method based study. It does not include details on how items will be summarised after a first or second expert rating and how they will be expressed in the final study table. I would recommend seeking expert advices on the methodology and statistical approach of a Delphi study.
--	--

REVIEWER	Sally Giles NIHR Greater Manchester Primary Care Patient Safety Translational Research Centre , University of Manchester, UK
REVIEW RETURNED	17-Mar-2017

GENERAL COMMENTS	On the whole this is a clearly written paper that focuses on a potentially interesting and relevant topic. I have some minor concerns about the paper, which are detailed below:  • In the abstract the authors refer to the “the problem of patient safety”. Do they mean error rates here, or patient safety more generally? This need clarification. There is also a typo under methods and analysis in priorities should read prioritise. • Introduction: This is currently sparse on references. The authors need to reference other relevant work. For example the sentence beginning with “A recent inquiry...” Needs a reference. There are several other similar examples where a reference is required. • The authors mention that developing countries need different strategies. Some further insight into what these might be would be useful. • Methods and analysis: in what way is the Delphi approach going to be modified? This isn’t clear in the manuscript. • The section on the use of the Delphi method in patient safety research could be simplified as currently it is quite repetitive. I think the main point here should be that Delphi has been used in patient safety research with examples. • Who are the experts? Will there be anyone from the Libyan healthcare system? A flow diagram illustrating the 3 stages of the Delphi process would help with clarity for this section. • I would be interested to gain some insight into how the authors think patient safety priorities will differ in developing countries. • I would like some more information on how the authors think the research will initiate collaborative research between developing and developed countries.
--

VERSION 1 – AUTHOR RESPONSE

Reviewer: 1

Reviewer Name: Marise Reis de Freitas

Institution and Country: Universidade Federal do Rio Grande do Norte, Brazil Competing Interests: None declared

This paper will contribute to research in developing countries.

Please in line 33 at abstract, you have to substitute the word hard for harm.

This has been done

Reviewer: 2

Reviewer Name: Helen Niemeyer (for Christine Knaevelsrud) Institution and Country: Freie Universität Berlin, Germany Competing Interests: None declared

The protocol under review deals with a very important topic and has several strengths. However, a number of aspects require a revision.

1. p.1, highlight 1 is unclear: Does "looks at research in Libya" mean that research conducted in Libya is analysed?

This has been paraphrased. It means that it focuses on the need to conduct more research about patient safety in Libya

2. The English language of the protocol needs to be revised, e.g. first sentence on p. 2.

Noted and the final version will be proof-read by English language editors

3. What is the meaning of the sentence "A recent inquiry in the United Kingdom ... workplace." on p. 2 in the context of patient safety?

This enquiry found a lot of issues that led to the occurrence of harm to patients. Illustrated further in the text.

4. Can the authors provide examples for the most important patient safety risks in developed countries?

These has been added which include hospital-acquired infections, patients being given the wrong drugs, adverse drug reactions, surgical complications and delays in diagnosis.

5. Will the study authors serve as oversight committee in the study?

Yes, this is a modified Delphi study, the first author will be acting as the main coordinator of this study supported by guidance from the other two co-authors. An ethical approval has been obtained from Imperial College Research Ethics Committee (ICREC: 16IC3598)

6. Does any of the authors have expertise on the Delphi method, e.g. serve as methodologist in former Delphi studies?

The main coordinator has received training in how to conduct Delphi studies. This study is a modified version of the Delphi technique. The two co-authors has extensive experience in health services research and governance

7. Prior to this project, did the authors conduct reviews of guidelines and the existing literature on patient safety (e.g. in developed countries)? The authors briefly refer to the SEIPS model. Please provide sufficient information about the model, whether it is related to developed or developing countries, and whether questions were derived from it for the Delphi process?

Yes, the first author's PhD is about improving patient safety in developing countries and have carried out extensive research on patient safety models which are implemented in developed countries. More information about the SEIPS model has been added.

8. Were questions for the Delphi process derived from the WHO papers that the authors refer to, which seem to be specifically about developing countries?

Yes, as well as other relevant literature. This is to focus on the main domains of patient safety such as

regulations and laws, training and education, monitoring and measurement, leadership and management etc..

9. Can the authors provide reasons why Libya was chosen?

Libya was chosen as it is the first author's country and his PhD research is about improving patient safety in Libya. It is also the fact that Libya is going through transition following the recent political changes since 2011. It is also located in North Africa which shares some of the contextual factors with other Arab and African countries.

10. Can the authors provide basic information about the current state of the Libyan health care system?

A brief overview has been added to the manuscript

11. One aim of the study (p.4) is to develop a model that is tailored to the Libyan context. Is this step supposed to be conducted by the international experts on patient safety? In that case, how do the authors make sure that their knowledge about Libya is sufficient?

No, this will be developed by the authors of the study based on the findings of the Delphi study.

12. Which time frame do the authors base their model for the implementation of patient safety structures in the Libyan health care system on (e.g. the next 10 years), in order to make it practically applicable?

The study does not aim to propose a timeline but would provide a guidance on how to improve patient safety based on lessons learnt from other developed nations. This will help policy makers in developing countries to focus on such elements and further invest to enhance the capabilities of their systems.

13. Can the authors provide detailed information on the web-based questionnaire they apply in the pre-Delphi process and on the pre-prepared identified domains and topics in the online questionnaire in round 1 (p. 5)?

The questionnaire for round one has been uploaded with this manuscript

14. Does the step that the panel members have to identify five barriers to implementation have anything to do with the domains in the online questionnaire? Do the members have to rank the five most important domains, or are domains and barriers two different topics?

The barriers are different to the domains.

The participants will be asked to rate the domains in round 1,2 and 2.

In round 1, the participants will be asked to identify five barriers that hinder patient safety systems implementation.

In round 2, the participants will be asked to rank the most important barriers that were identified in round 1.

15. How are the experts defined? Can the authors provide more information on their profession and scientific background (p.6)?

The experts are selected based on their international work on patient safety and recent publications in the field. The experts are also selected based on their familiarity with health systems in developing countries such as those working the WHO EMRO region. Added some notes in the manuscript to highlight this.

Reviewer: 3

Reviewer Name: Professor Salman Rawaf

Institution and Country: Imperial College London, UK Competing Interests: I know the three authors as they work with me in the same department. However, I am not involved in this research (PhD Research topic and not Departmental Research)

This is an important protocol on topic of great interest and implications in developing countries. While the flow of the paper is good I believe the linguistic issues have to be addressed before any consideration for publication (please see my comments highlighted on the attached manuscript).

The comments highlighted in the attached manuscript were taken into consideration and revised accordingly.

Reviewer: 4

Reviewer Name: A-Prof Guy Haller

Institution and Country: Geneva University Hospitals-Switzerland Competing Interests: None

This protocol describes an expert consensus gaining study based on the Delphi methodology. It aims at identifying key factors that are the most important in the improvement of patient safety and at highlighting barriers that may hinder the implementation of these factors in developing countries. Libya will be used as a case-study. The study protocol plans to recruit experts in Africa, the World Health Organization, the United States, the United Kingdom and Arab countries. Such studies are important to the overall advancement of the patient safety agenda across countries and more specifically, in developing countries.

While the study is important and the protocol includes detailed information regarding study aim and gap in knowledge, the lack of a clear and comprehensive methodology section in the manuscript hinders the protocol to reach its goal.

Additional details are needed at this stage. First, there appears to be some tension between the concept of an extensive Delphi process aimed at identifying needs in developing countries and the choice of Libya as a case-study. It is unclear how factors identified on Libya could be transposed to, for instance, Guatemala, Cambodia or the Philippines, all being developing countries. The generalizability issue is substantial in this study. Secondly, a Delphi study is not a qualitative study. As a result, detailed information and rating sheet that will be sent to expert should be provided. This sheet should include all factors and aspects on which expert opinion will be sought. Thirdly, it is unclear how a highly selected group of experts in Africa, the World Health Organization, the United States, the United Kingdom and Arab countries are representative of all developing countries. What about Asia, South America, the Pacific islands? Finally, the statistical method provided in the protocol does not relate to a Delphi method based study. It does not include details on how items will be summarised after a first or second expert rating and how they will be expressed in the final study table.

I would recommend seeking expert advices on the methodology and statistical approach of a Delphi study.

Thank you for your all comments. These have been taken into consideration. In the final report of this Delphi study, the limitations section will highlight some of the points raised above. To clarify, the questionnaire for the first round has been included to show the rating system used. This is a modified version of the Delphi methodology and only descriptive analysis will be needed to identify the main priorities. The authors believe that the participants would mainly focus on the important factors that can help in improving patient safety in developing countries in general while focusing on a country-specific system to help them guide their thoughts and views.

In regards to the generalisability of the study. The authors believe that health systems in developing countries share some of the weaknesses such as the absence of strict regulatory frameworks, lack of staff, lack of standardised processes and ineffective monitoring etc.. therefore, the study does not imply that what work in Libya would work in other developing countries but we believe that such factors that are identified to Libya would also be of relevance to other countries. These might include the need to improve the teaching curriculum within medical and nursing colleges/schools, investments in information technology, establishing independent regulators etc..

Reviewer: 5

Reviewer Name: Sally Giles

Institution and Country: NIHR Greater Manchester Primary Care Patient Safety Translational Research Centre, University of Manchester, UK Competing Interests: None

On the whole this is a clearly written paper that focuses on a potentially interesting and relevant topic. I have some minor concerns about the paper, which are detailed below:

1. In the abstract the authors refer to the “the problem of patient safety”. Do they mean error rates here, or patient safety more generally? This needs clarification. There is also a typo under methods and analysis in priorities should read prioritise.

Patient safety more generally as reported in many WHO reports, they tend to describe the issue of patient safety as a major public health problem

2. Introduction: This is currently sparse on references. The authors need to reference other relevant work. For example the sentence beginning with “A recent inquiry....” Needs a reference. There are several other similar examples where a reference is required.

The reference has been added.

3. The authors mention that developing countries need different strategies. Some further insight into what these might be would be useful.

A new sentence added to clarify this.

4. Methods and analysis: in what way is the Delphi approach going to be modified? This isn't clear in the manuscript.

Modified in terms of the number of participants and the approach to the first-round survey development. Clarified within the manuscript

5. The section on the use of the Delphi method in patient safety research could be simplified as currently it is quite repetitive. I think the main point here should be that Delphi has been used in patient safety research with examples.

We think that this would help in further validating the use of Delphi in patient safety research that is why we have added few references there. It was also meant to show that different types of Delphi processes are adopted such as modified or large studies. We will consider revising if this response does not justify our use of more than one example in this section.

6. Who are the experts? Will there be anyone from the Libyan healthcare system? A flow diagram illustrating the 3 stages of the Delphi process would help with clarity for this section.

More information added about the experts. A diagram has been added (Figure 1)

7. I would be interested to gain some insight into how the authors think patient safety priorities will differ in developing countries.

More insights added in the discussion about the needs of developing countries, lack of regulatory frameworks and governance systems, infrastructural needs

8. I would like some more information on how the authors think the research will initiate collaborative research between developing and developed countries.

Added a new sentence to point out the importance of bridging the gap between developed and developing countries.

VERSION 2 – REVIEW

REVIEWER	Helen Niemeyer Freie Universität Berlin Germany
REVIEW RETURNED	24-May-2017

GENERAL COMMENTS	The manuscript has improved significantly and I have no further comments.
---